# Coupled Genome-Wide DNA Methylation and Transcription Analysis Identified Rich Biomarkers and Drug Targets in Triple-Negative Breast Cancer

**DOI:** 10.3390/cancers11111724

**Published:** 2019-11-04

**Authors:** Maoni Guo, Siddharth Sinha, San Ming Wang

**Affiliations:** Cancer Centre and Institute of Translational Medicine, Faculty of Health Sciences, University of Macau, Macau 999078, China; yb87627@um.edu.mo (M.G.); siddharths@um.edu.mo (S.S.)

**Keywords:** TNBC, DNA methylation, gene expression, therapeutic targets

## Abstract

Triple-negative breast cancer (TNBC) has poor clinical prognosis. Lack of TNBC-specific biomarkers prevents active clinical intervention. We reasoned that TNBC must have its specific signature due to the lack of three key receptors to distinguish TNBC from other types of breast cancer. We also reasoned that coupling methylation and gene expression as a single unit may increase the specificity for the detected TNBC signatures. We further reasoned that choosing the proper controls may be critical to increasing the sensitivity to identify TNBC-specific signatures. Furthermore, we also considered that specific drugs could target the detected TNBC-specific signatures. We developed a system to identify potential TNBC signatures. It consisted of (1) coupling methylation and expression changes in TNBC to identify the methylation-regulated signature genes for TNBC; (2) using TPBC (triple-positive breast cancer) as the control to detect TNBC-specific signature genes; (3) searching in the drug database to identify those targeting TNBC signature genes. Using this system, we identified 114 genes with both altered methylation and expression, and 356 existing drugs targeting 10 of the 114 genes. Through docking and molecular dynamics simulation, we determined the structural basis between sapropterin, a drug used in the treatment of tetrahydrobiopterin deficiency, and *PTGS2*, a TNBC signature gene involved in the conversion of arachidonic acid to prostaglandins. Our study reveals the existence of rich TNBC-specific signatures, and many can be drug target and biomarker candidates for clinical applications.

## 1. Introduction

Triple-negative breast cancer (TNBC) is a subtype of breast cancer, accounting for 15–20% of all breast cancer cases. It is characterized by the absence of estrogen receptor (ER), progesterone receptor (PR), and epidermal growth factor receptor 2 (HER2/neu) [1,2], the three receptors targeted by hormone-based effective therapies. TNBC often occurs in young-age patients, and it is highly aggressive with early relapse, metastasis, and poor prognosis. Genetic predisposition, such as germline mutations in *BRCA1*, was determined as a contributing factor for TNBC [3]. Despite efforts made so far, progress remains slow in clinical TNBC treatment, largely due to the lack of TNBC-specific biomarkers for effective intervention [4,5,6].

DNA methylation and gene expression are promising sources to identify TNBC-specific biomarkers. For example, studies observed that altered DNA demethylation in *TET1* can affect the PI3K/mTOR pathway in TNBC [7], alternated DNA methylation in *DNMT1* is associated with higher histologic grade, ER negativity, PR negativity, and higher Ki-67 in TNBC [8], hypermethylation in *DUSP1* is linked with increased risk of TNBC [9], and methylation in *BRCA1* promoter has poor survival in TNBC patients [10]. Multiple alternatively expressed genes were also identified in TNBC [11,12]. For instance, overexpression of *SPHK1* may increase spontaneous metastasis to lungs in TNBC patients [13], high expression of *EN1* is associated with poor survival and increased risk of developing brain metastasis in TNBC patients [14], and *WWOX* may suppress TNBC cell metastasis and proliferation by downregulating the JAK2/STAT3 pathway [15]. However, the clinical impact from these studies remains limited, either due to their strict representatives in TNBC or due to the lack of drugs targeting these potential biomarkers.

We reasoned that (1) the lack of three key receptors in TNBC should lead to its unique signature distinguishable from other types of breast cancer; for example, lack of ER may cause substantial changes in gene expression in TNBC [16]; (2) as methylation directly controls gene expression, focusing on the DNA methylation-regulated genes as one unit may substantially increase the specificity of the detected TNBC signature from non-TNBC types of breast cancer; (3) choosing proper control could be critical to increasing the sensitivity in identifying TNBC-specific signatures, which might not be so remarkable from other types of breast cancer with different combinations of the three receptor statuses; and (4) specific drugs may target the detected TNBC-specific signatures. Based on these considerations, we designed a system to identify a TNBC-specific signature. In this system, we used triple-positive breast cancer (TPBC) as the control to gather genome-wide methylation and gene expression data to locate the altered methylations coupled with altered expression of the same genes. Potential existing drugs targeting the identified TNBC signature genes were also identified. Applying this approach, we identified a total of 114 genes considered as TNBC signatures, and 356 existing drugs potentially targeting 10 of the 114 genes. Using a molecular dynamics simulation, we predicted the structural basis for the interaction between a TNBC-signature gene, *PTGS2* (also known as *COX-2*), and an existing drug, sapropterin, targeting *PTGS2*.

## 2. Materials and Methods

### 2.1. Sample Datasets Used in the Study

We used 116 TNBC cases and 100 TPBC cases from The Cancer Genome Atlas (TCGA) data. We downloaded both methylation (Illumina Infinium HumanMethylation450 BeadChip) and expression data (RNA-SeqV2) generated by TCGA from UCSC xena (https://xenabrowser.net/hub/) [17] (Appendix A), for which 84 TNBCs and 64 TPBCs contained both DNA methylation and gene expression information (Appendix A). We also used an independent DNA methylation dataset (GSE78758) [11], from the GEO database, containing 35 TNBCs from GSE78751 and 70 TNBCs from GSE78754, and an expression dataset from cBioPortal resource including 299 TNBCs [18].

### 2.2. Analysis of DNA Methylation Data

The Illumina HumanMethylation450 BeadChip array contains 485,577 probes covering 99% (*n* = 21,231) of RefSeq genes. The raw methylation intensities for each probe were represented as β-values (Equation (1)), which were converted into M-values with the beta2m function in R package lumi for statistics analysis [19,20]. 5’-C-phosphate-G-3’ (CpG) methylation data between TNBC and TPBC were compared with R package limma to identify differentially methylated CpG sites (DMSs). Each *p*-value was adjusted as false discovery rate (FDR) using the Benjamini and Hochberg (BH) method. The threshold for DMS identification was an adjusted *p*-value < 0.05 and an absolute delta Β-value >0.2 (0 < β-value < 1; CpG delta β-value: Equation (2)).

CpG site and gene mapping files were downloaded from https://www.illumina.com/. The average β-values for each region (TSS1500, TSS200, 5’-UTR—5′-untranslated region, first exon, gene body, 3’-UTR—3′-untranslated region, and intergenic region, TSS—transcriptional start site) were calculated according to all CpG sites at the corresponding region (Equation (3)), and the average β-value was converted to an M-value with the beta2m function. Average regional methylation data between TNBC and TPBC were compared with R package limma to identify differentially methylated regions (DMRs). We identified hypermethylated DMRs with a threshold of adjusted *p*-value < 0.05 and a delta β-value >0.2, and hypomethylated DMRs with a threshold of adjusted *p*-value < 0.05 and a delta β-value < −0.2 (region delta β-value: Equation (4)). Genes harboring DMRs were termed as differentially methylated genes (DMGs). The equations described above are listed below.
(1)$$\beta =\frac{M}{M+U+100},$$
(2)$$\Delta \beta =\beta_{tnbc}-\beta_{tpbc},$$
(3)$$\beta_{region}=2^{\left(\overset{k}{\underset{i=1}{\sum }}log2\left(\beta_{i}\right)\right)/k},$$
(4)$$\Delta \beta_{region}=\beta_{tnbcRegion}-\beta_{tpbcRegion},$$
where *M* is the methylated allele intensity, *U* is the unmethylated allele intensity, *i* = 1, 2, 3, …, *k*, and *k* is the number of CpG cites in a region.

### 2.3. Analysis of Gene Expression Data

Differential expression between TNBC (*N* = 84) and TPBC (*N* = 64) samples was analyzed with the R limma package. We adjusted each *p*-value as false discovery rate (FDR) using the Benjamini and Hochberg (BH) method. We used the log-transformed expression value to identify differentially expressed genes (DEGs), including upregulated genes with an adjusted *p*-value < 0.05 and logFC (fold change) > 2, and downregulated genes with an adjusted *p*-value < 0.05 and logFC < −2 in TNBC compared with TPBC.

### 2.4. Analysis of DMGs and DEGs in Different Regions

To test the relationship between methylation and expression, we calculated the intersection of DMGs and DEGs as differentially methylated and expressed genes (DMEGs), and classified them into four distinct groups: HypoUp, HypoDown, HyperUp, and HyperDown (Table 1, Appendix A). We determined the proportion of different groups in DMEG-enriched regions and extracted the DMSs within different regions of all DMEGs. 

### 2.5. Functional Enrichment Analysis

We applied enrichGO and enrichKEGG functions in Bioconductor package clusterProfiler for DMGs and DEGs to explore important processes or pathways involved in TNBC genesis and development [21]. We performed Gene Ontology (GO) and Kyoto Encyclopedia of Genes and Genomes (KEGG) annotation using the Database for Annotation, Visualization and Integrated Discovery (DAVID) functional annotation tools (http://david.abcc.ncifcrf.gov/, *version* 6.8) [22], with a *p*-value < 0.05 considered as statistically significant in both clusterProfiler and DAVID analyses.

### 2.6. Evaluation of Methylation and Expression Biomarkers

We used principal component analysis for all DMSs in DMEGs to distinguish between TNBC and TPBC. Two random forest classifiers were trained based on the methylation level of all DMSs and expression level of all DMEGs using the R randomForest package. The leave-one-out cross-validation (LOOCV) method was used to evaluate the performance of the classifier. Plots of the receiver operating characteristic (ROC) curve of the classifier and the calculation of the area under the curve (AUC) were fulfilled using the R verification package. We performed stratification analysis to avoid the interferences from other clinical factors including age, menopause status, race, and stage.

### 2.7. Identification of Potential Drug Targets

We screened DrugBank to identify drugs potentially targeting the upregulated DMEGs. DrugBank (*version* 5.1.2, released 20 December, 2018) contains 12,112 drug entries including 2556 Food and Drug Administration (FDA)-approved small-molecule drugs, 1280 FDA-approved biotech (protein/peptide) drugs, 130 nutraceuticals, and 5842 experimental drugs [23]. We downloaded the drug–target interactions and searched for the drugs targeting DMEGs to identify those targeting upregulated DMEGs in TNBC.

To explore the determination of structural basis between protein targets and drug-binding sites, we downloaded protein structure 5IKV for gene *PTGS2* (*COX-2*) from the Protein Data Bank (PDB) (www.rcsb.org) and optimized its structure for binding calculation using UCSF chimera [24]. FDA-approved drugs targeting *PTGS2* were screened for their ability to bind to 5IKV using molecular docking and molecular dynamics simulations (MDS). Binding pattern determination for the prostaglandin-endoperoxide synthase 2 (PTGS2)–inhibitor complex was performed using Autodock 4.0 [25]; MDS of the active site for the PTGS2–inhibitor complex was carried out using the Gromacs 5.1 suite of programs with gromos force field [26] (see details in Supplementary Methods).

## 3. Results

### 3.1. Differentially Methylated Genes (DMGs) in TNBC

To identify DMRs in TNBC, we extracted the DNA methylation data from 84 TNBCs and 64 TPBCs and performed comparative analysis. In this study, we mainly focused on three genomic regions: TSS1500, TSS20, and gene body. In total, we identified 5345 statistically significant (adjusted *p*-value < 0.05) different DMRs between TNBC and TPBC, including 396 located in TSS1500, 394 in TSS200, 270 in gene body, and 2063 in intergenic regions (Figure 1A–C; Appendix A). We classified these DMRs into hypermethylated and hypomethylated groups (see Section 2). We identified 911 DMGs with 273 hypermethylated and 638 hypomethylated ones, including 104 hypermethylated DMRs and 292 hypomethylated DMRs in the TSS1500 region, 95 hypermethylated DMRs and 299 hypomethylated DMRs in the TSS200 region, and 99 hypermethylated DMRs and 171 hypomethylated DMRs in the gene body region (Figure 1D,F). The number of hypomethylated DMRs was significantly (Student’s *t*-test: *p*-value = 0.06) larger than that of hypermethylated ones, indicating that TNBC is less methylated than TPBC. Of all the DMGs harboring DMRs, only 1.65% (15/911) had methylation alterations in all three regions, suggesting that methylation is region-specific in TNBC (Figure 1E). The unsupervised hierarchical clustering analysis showed that DMGs could effectively distinguish TNBC from TPBC in four regions (Appendix A). Functional enrichment analysis revealed that these DMGs participated in important biological processes and pathways, such as extracellular matrix organization, cytokine–cytokine receptor interaction, cell adhesion molecules, and the TNF signaling pathway (Figure 1G,H).

### 3.2. Differentially Expressed Genes (DEGs) in TNBC

To identify DEGs in TNBC, we firstly extracted the gene expression data of 84 TNBCs and 64 TPBCs and performed comparative analysis. In total, we identified 710 statistically significant (adjusted *p*-value < 0.05) DEGs including 332 upregulated and 378 downregulated genes (Figure 2A; Appendix A). Remarkably, the three receptor coding genes, *ESR1*, *PGR*, and *HER2*/*neu*, were significantly downregulated in TNBC compared with TPBC (Figure 2B–D) [27]. Of the upregulated DEGs, many are known to be associated with TNBC, such as *FABP7*, *GABRP*, and *VGLL1* (Figure 2E–G) [28,29,30]. Unsupervised hierarchical clustering analysis revealed that these DEGs effectively distinguished between TNBC and TPBC (Figure 2H). Function analysis showed that the estrogen signaling pathway, one of the most significantly enriched functional pathways, harbored important TNBC-associated genes, including *TFF1*, *ESR1*, *PGR*, *EGFR*, *KRT5*, *KRT14*, and *KRT17* (Figure 2I). Same analysis in 332 upregulated and 378 downregulated DEGs showed that they were involved in TNBC-related biological processes, such as hormone transport, extracellular matrix, and epidermis development (Appendix A). 

### 3.3. Differentially Methylated and Expressed Genes (DMEGs) in TNBC

We analyzed the relationship between methylation and gene expression by integrating DMGs and DEGs in three genomic regions: TSS1500, TSS200, and gene body. We identified a total of 114 DMEGs, including 86 in only one of the three regions and 28 in two of the three regions (Figure 3A–C; Appendix A), and classified these into four different groups (HypoUp, HyperUp, HyperDown and HypoDown) (Figure 3D–F; Appendix A). The HypoUp group was the most common group, accounting for 61%, 71%, and 44% in the three regions, respectively (Figure 3G–I). The HyperDown group occupied the second most common position in TSS1500 and TSS200 regions. No significant HyperDown correlation was observed in the gene body region, reflecting its insignificant role of methylation in gene expression regulation. In total, we identified 114 DMEGs, in which *GABRP* was the most upregulated gene, and *TFF1* was the most downregulated gene in both TSS1500 and TSS200 regions in TNBC.

### 3.4. DMEGs Predicting TNBCs

Our coupled analysis identified 114 DMEGs containing 250 DMSs distributed in TSS1500 and TSS200 regions. The circos plot showed that the 114 DMEGs were distributed across the whole genome except for chromosomes X and Y (Figure 4A). To further explore the differences between TNBC and TPBC in DNA methylation and gene expression, we constructed 250-DMS- and 114-DMEG-based random forest classifiers, and performed principal component analysis (PCA) and ROC analyses in TNBC and TPBC. The results showed that nearly all samples were correctly classified into their groups (Figure 4B,C) with high significance (Figure 4D,E)(*p* = 1.66 × 10^−23^, AUC = 0.977, Figure 4D for the 250-DMS classifier; *p* = 1.83 × 10^−24^, AUC = 0.987, Figure 4E for the 114-DMEG classifier), confirming the existence of differential methylation and expression between TNBC and TPBC. Using the 250-DMS classifier, we tested in 105 TNBCs from the GSE78751 and GSE78754 subsets (see Section 2) and observed high accuracy of TNBC classification (96/105, 91.4%); using the 114-DMEG classifier, we tested in 299 TNBCs from the cBioPortal database and also observed high accuracy of TNBC classification (286/299, 95.6%).

To decipher the functional categories of the 114 DMEGs, we performed DAVID annotation and classified these into biological process (BP), cellular component (CC), and molecular function (MF) terms, and KEGG pathways. The significantly enriched categories included cell proliferation, immune response, cell differentiation, extracellular space, sequence-specific DNA binding, and transcriptional mis-regulation in cancer (Appendix A). The GO terms “cell proliferation” and “cell differentiation” were significantly enriched by several TNBC-associated genes, including *CXCL1*, *ELF5*, *LIPG*, and *UCHL1*, which are known to play crucial roles in the initiation and metastasis of breast cancer and as potential TNBC diagnostic markers and therapeutic targets [31,32,33,34]. Genes involved in immune response were also common in DMEGs, implying that some immune-related DMEGs can be potential therapeutic targets for TNBC [35]. 

We conducted stratification analysis for other clinical factors, including age, stage, menopause, and race, to know if these factors could also contribute to DMEGs. The results showed that the 114-DMEG predictor performed similarly in age (*p* = 3.12 × 10^−16^, AUC = 0.992 for age >50 group; *p* = 7.37 × 10^−10^, AUC = 0.979 for age ≤50 group), menopause (*p* = 4.67 × 10^−16^, AUC = 0.988 for post-menopause group; *p* = 1.68 × 10^−5^ AUC = 0.946 for pre-menopause group), race (*p* = 1.29 × 10^−16^, AUC = 0.985 for White group; *p* =1.1 × 10^−5^ AUC = 1 for Black group), and tumor stage (*p* = 2.04× 10^−6^ AUC = 1 for stage I group; *p* = 9.5 × 10^−14^ AUC = 0.977 for stage II group; *p* = 2.39 × 10^−7^ AUC = 0.972 for stages III and IV group) (Appendix A).The results indicated that the 114 DMEGs were independent of these factors.

### 3.5. Multiple DMEGs are Potential Druggable Targets

The DrugBank database contains available FDA-approved drugs and experimental compounds, and provides detailed drug and drug target information. To explore if there could be any drugs targeting the DMEGs, we allocated the 114 DMEGs into several categories, including receptor, protein, and other structural components (Table 2). We searched the 75 upregulated DMEGs in the DrugBank database and identified 356 therapeutic drugs targeting 10 of the 75 upregulated DMEGs, in which nine harbored HypoUp DMRs, and one harbored HyperUp DMRs (Table 3; Appendix A). Among the 356 drugs targeting the 10 DMEGs, 288 were Food and Drug Administration (FDA)-approved drugs for the treatment of different diseases (Table 3; Appendix A).

Among the 10 genes in DMEGs with targeted drugs, several are known to be associated with TNBC, such as *PTGS2*, *GSTP1*, *GABRP*, and *TF* (Table 3). *PTGS2*, also known as *COX-2*, harbored 14 CpG sites (nine DMSs) in its TSS500, TSS200, and gene body regions, targeted by 111 drugs. *PTGS2* is overexpressed in TNBC, and promotes metastasis by regulating crucial molecules (Bcl-2, MRP4, PGT, 15-PGDH, TGFβ, etc.) involved in multiple tumor-promoting signaling pathways [36,37,38]. *GSTP1*, harboring six CpG sites (four DMSs) in the gene body region was targeted by 37 drugs. *GSTP1* is a driver gene and potential prognostic and therapeutic marker for TNBC [39,40]. *GABRP* harboring six CpG sites (three DMSs) in its TSS1500 and TSS200 regions is closely related with basal-like TNBC and is a prognostic marker for TNBC [28,41]. *GABRP* was targeted by 16 drugs. *TF* harbors seven CpG sites (three DMSs) in its TSS1500 and TSS200 regions, and is known to play key roles in drug delivery in TNBC tumor cells [42,43]. *TF* was targeted by 32 drugs. In addition, S100B can suppress the migratory capacity of ER-negative breast cancer cells, and the expression of S100B is a predictive marker for metastasis in breast cancer [44,45]. The muscarinic acetylcholine receptor (CHRM3), although unknown to be related to TNBC, is a key factor in the development of prostate cancer, bladder cancer, and endometrial carcinoma [46,47,48].

### 3.6. Structural Basis of Sapropterin Bound to PTGS2

The 111 drugs targeting PTGS2 included 82 FDA-approved drugs, three anti-neoplastic drugs, and 19 potential drugs (Figure 5A). We selected PTGS2 (PDB identifier (ID): 5IKV) and sapropterin as an example to elucidate the structural basis between the TNBC signature genes and the targeting drugs. The active site of PTGS2 comprises Arg-513, Phe-518, Gly-526, Leu-352, Val-349, His-90, and Gln-192 [49]. A total of 11 drugs were identified on the basis of the binding energy and the number of hydrogen bonds with the active site residues (Table 4). The binding energy was in the range from −6.6 kcal/moL to −3.19 kcal/moL; the binding energy for sapropterin was −6.15 kcal/moL with a total of six hydrogen bonds, three with Asn-87 and one each with His-90, Lys-511, and Glu-520 (Figure 5B). 

Based on the docking results, the 5IKV–sapropterin conformation was used for MDS analysis (Table 4). In MDS, RMSD (root-mean-square deviation) revealed that the native and the drug-bound 5IKV demonstrated similar convergence in the equilibration phase for 0.5 ns, thereafter showing varying mobility in the active site (Figure 5C). The bound complex showed a lower degree of convergence around 0.2–0.25 nm compared to the native protein with a convergence of 0.3–0.4 nm. RMSF (root-mean-square fluctuation) analysis showed that the bound complex had a higher degree of resilience at residue positions Asn-87, His-90, Lys-511, and Glu-520 than the native protein (Figure 5D). Rg (radius of gyration) analysis showed that sapropterin-bound PTGS2 lost the compactness of the native protein structure (Figure 5E). NH-bond (number of hydrogen bonds) analysis showed that the bound complex had a greater number of hydrogen bonds (~580–590) compared with the native PTGS2 structure (~400–410), resulting in a more compact complex (Figure 5F). The results from docking and MDS analyses showed the high binding affinity and stable interaction between PTGS2 and sapropterin.

## 4. Discussion

Abnormal gene expression contributes to cancer development. This is well reflected in TNBC, in which *ER*, *PR*, and *HER2*/*neu* are not expressed [50,51,52]. Furthermore, abnormal gene expression in cancer is largely caused by abnormal methylation in regulatory regions. Therefore, abnormal methylation and gene expression provide attractive sources to identify biomarkers and drug targets for TNBC. Despite many efforts made in mining the sources, information for this purpose remains limited [53,54,55]. 

Two improvements were used in our study in order to improve the situation. One is that we used TPBC instead of other types of breast cancer as the control in order to provide the highest contrast to TNBC. We reasoned that the difficulty in identifying TNBC-specific signals could be caused by the marginal differences in abnormal signatures between TNBC and non-TNBC. Hence, increasing the signal-to-noise ratio between TNBC and non-TNBC would be critical to identify the TNBC-specific signatures. The other is that we coupled methylation sites and their located genes as a single unit. We reasoned that this would significantly decrease the noise from unrelated methylation and gene expression, leading to increased specificity to identify TNBC-specific signatures. Indeed, our study following these steps led to identifying 114 TNBC-specific DMEGs, and 356 potential therapeutic drugs targeting 10 of the 114 DMEGs, of which 288 were existing drugs used for the treatment of various diseases including cancer, indicating that our approach indeed provides a powerful means to identify TNBC-specific signatures.

The structural relationship between PTGS2 and sapropterin is interesting. PTGS2 is an enzyme that converts arachidonic acid to prostaglandins. Animal studies showed that overexpression of the *PTGS2* in mammary glands can induce tumorigenesis, and that *PTGS2* is often over-expressed in human breast cancer [56,57]. Sapropterin is a drug used in the treatment of tetrahydrobiopterin (BH4) deficiency through its specific inhibition of PTGS2; however, it is not used for the treatment of any specific type of cancer. Our structural analysis revealed the structural basis of the PTGS2 and sapropterin interaction to support the potential of using sapropterin to treat TNBC through its specific targeting of PTGS2. Further studies are needed to explore PTGS2-specific inhibitors for TNBC treatment. For 65 of the 75 upregulated DMEGs without existing therapeutic drugs in the current drug database, they can be potential targets for new TNBC-targeted drug development, and as clinical biomarkers for TNBC diagnosis and prognosis.

In summary, data from our study demonstrate the presence of rich biomarkers in TNBC and their potential as drug targets and biomarkers for clinical TNBC application.

## 5. Conclusions

We highlighted the crucial roles of DMGs and DEGs in the development of TNBC. Our study demonstrated 114 dysregulated DMEGs in TNBC, which could serve as molecular biomarkers for the early detection of TNBC. Additionally, we identified 356 potential drugs targeting 10 of the 114 genes, of which 288 are FDA-approved drugs, and some are applied to clinical cancer treatment.

## Figures and Tables

**Figure 1 cancers-11-01724-f001:**
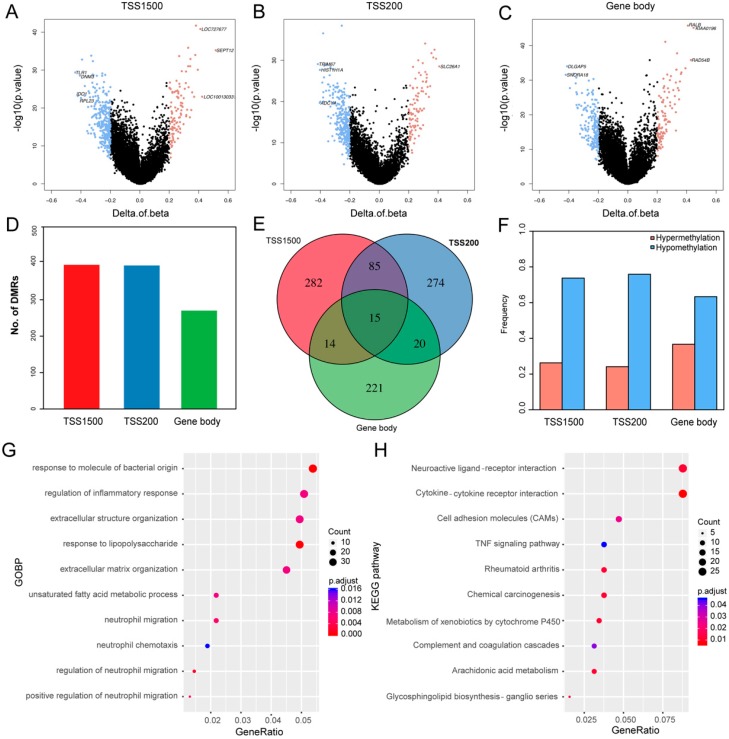
Differentially methylated genes (DMGs) in triple-negative breast cancer (TNBC). (**A**–**C**) Volcano plots showing the distributions of DMGs in TSS1500, TSS200, and gene body regions. The red and blue dots represent the significantly hyper- and hypomethylated DMGs. (**D**) Bar plot showing the numbers of DMGs in TSS1500 (*n* = 396), TSS200 (*n* = 394), and gene body (*n* = 270) regions. (**E**) Venn plot showing the intersections of DMGs between the three regions. Only 15 genes were differentially methylated in all three regions, although there were numerous region-specific DMGs in TSS1500 (*n* = 282), TSS200 (*n* = 274), and gene body (*n* = 221). (**F**) Bar plots showing the proportions of methylation in the three regions. (**G**) The top 10 Gene Ontology Biological Process (GOBP) terms in the three regions. (**H**) The top 10 Kyoto Encyclopedia of Genes and Genomes (KEGG) pathways in the three regions.

**Figure 2 cancers-11-01724-f002:**
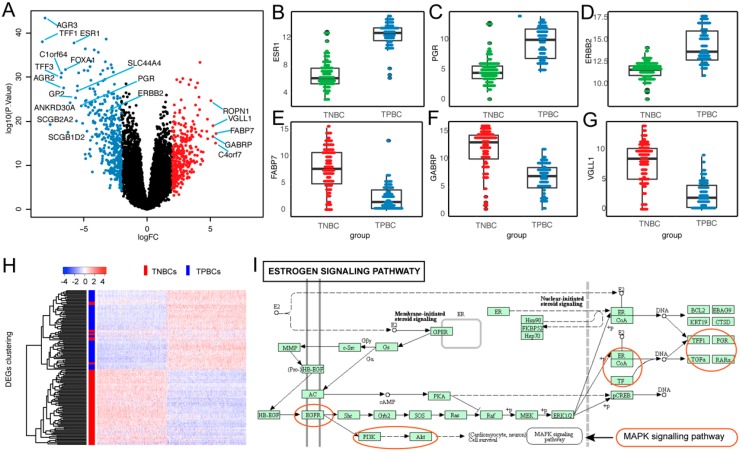
Differentially expressed genes (DEGs) in TNBC. (**A**) Volcano plot showing all 710 DEGs in TNBC. The red and blue dots represent the significantly up- and downregulated DEGs. (**B**–**G**) Box plots showing the distributions of downregulated *ESR1*, *PGR*, and *HER2*/*neu* genes and upregulated *FABP7*, *GABRP*, and *VGLL1* genes in TNBC tumors (*n* = 84), compared with triple-positive breast cancer (TPBC) tumors (*n* = 64). (**H**) Heatmap showing the expression profile of 84 TNBCs and 64 TPBCs across 710 DEGs. (**I**) Significantly enriched DEGs in estrogen signaling pathway.

**Figure 3 cancers-11-01724-f003:**
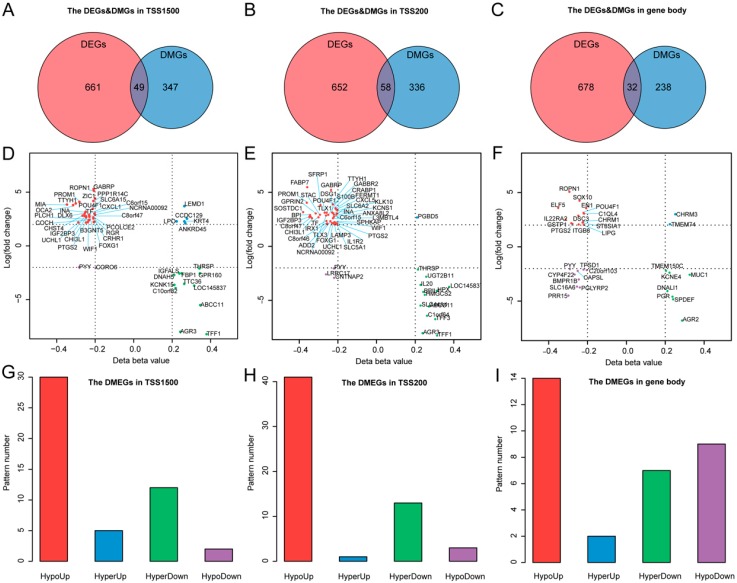
Differentially methylated and expressed genes (DMEGs) in TNBC. (**A**–**C**) Venn plots showing the DMEGs between DMGs and DEGs in TSS1500, TSS200, and gene body regions. (**D**–**F**) Quadrant plot showing DNA methylation and gene expression of DMEGs in TSS1500, TSS200, and gene body regions. The *x*-axis represents delta of mean methylation level (β-value) of all 5’-C-phosphate-G-3’ (CpG) cites with ≥20% difference in methylation for each gene between TNBC and TPBC. The *y*-axis represents log2-transformed fold change of gene expression between TNBC and TPBC. Vertical dashed lines indicate the threshold of 20% methylation difference, and horizontal dashed lines indicate the threshold corresponding to 2-logFC gene expression change. The four quadrants represent four methylation and expression patterns in TNBC: HypoUp, HyperUp, HyperDown, and HypoDown (see text for details). (**G**–**I**) Bar plots showing the number of four regulation patterns between methylation and expression of TNBC in three regions.

**Figure 4 cancers-11-01724-f004:**
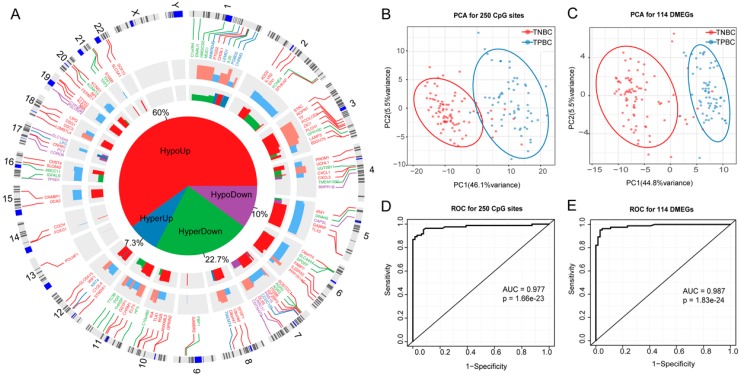
Prediction of TNBCs by DNA methylation and gene expression profiles. (**A**) Circos plot showing genome-wide locations of 114 DMEGs and their methylation and expression distribution. From outside to inside, the first circle shows chromosome distribution, the second shows the distribution of 114 DMEGs, the third shows the significance of differentially methylated CpG sites (DMSs), and the fourth shows the significance of DEMGs. The pie plot in the middle shows the proportion of four regulation patterns of HypoUp, HyperUp, HyperDown, and HypoDown between methylation and expression. (**B**–**C**) PCA plots for 84 TNBCs and 64 TPBCs by the 250-DMS and 114-DMEG predictors. (**D**) A predictor based on methylation values of 250 DMSs assigns all samples into TNBC and TPBC with high accuracy, with an AUC (area under the ROC—receiver operating characteristic curve) of 0.977 by ROC analysis. (**E**) A predictor based on expression values of 114 DMEGs assigns all samples into TNBC and TPBC with high accuracy, with an AUC (area under the ROC curve) of 0.987 by ROC analysis.

**Figure 5 cancers-11-01724-f005:**
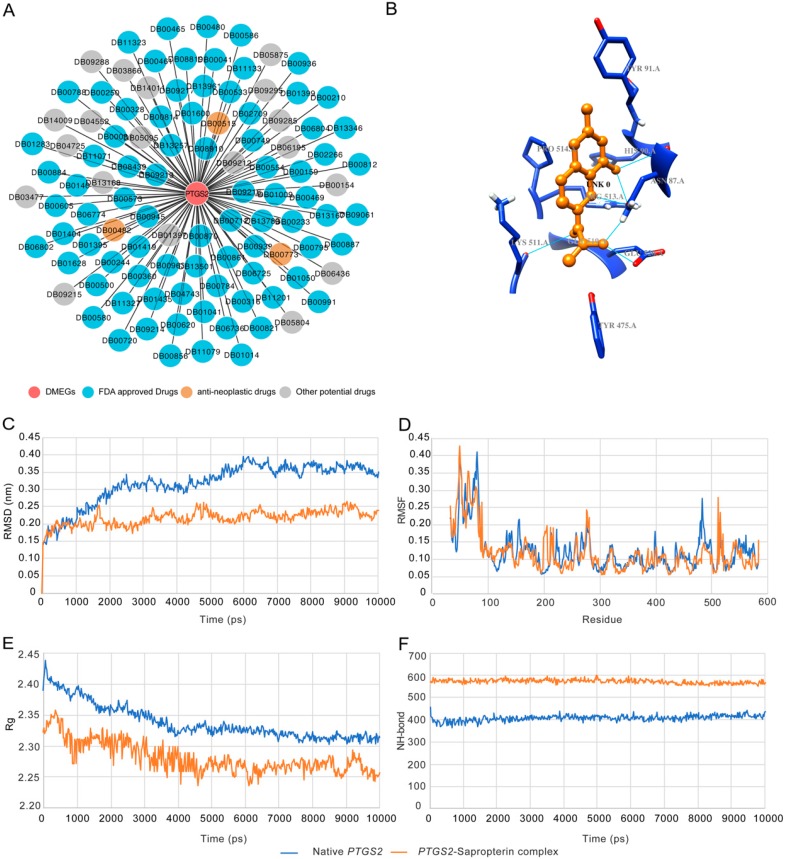
Example of PTGS2–sapropterin complex based on molecular dynamics simulation (MDS) analysis. (**A**) Drugs targeting PTGS2 predicted with our pipeline. (**B**) The three-dimensional (3D) structure of bound PTGS2–sapropterin complex. (**C**–**F**) The root-mean-square deviation (RMSD) profile, root-mean-square fluctuation (RMSF) profile, radius of gyration (Rg) profile, and the number of hydrogen bonds of native and bound PTGS2–sapropterin complex for a time period of 10 ns.

**Table 1 cancers-11-01724-t001:** The measurable cutoff of four patterns.

Groups	Methylation Cut-Off	Expression Cut-Off
HypoUp	adjusted *p*-value < 0.05 and delta β-value < −0.2	adjusted *p* value <0.05 and logFC > 2
HypoDown	adjusted *p*-value < 0.05 and delta β-value < −0.2	adjusted *p* value <0.05 and logFC < −2
HyperUp	adjusted *p*-value < 0.05 and delta β-value > 0.2	adjusted *p* value <0.05 and logFC > 2
HyperDown	adjusted *p*-value < 0.05 and delta β-value > 0.2	adjusted *p* value <0.05 and logFC < −2

**Table 2 cancers-11-01724-t002:** Allocated categories of 114 differentially methylated and expressed genes (DMEGs) for triple-negative breast cancer (TNBC).

Type	RefGene
Receptors	*GPR160, GABRP, RGR, CRHR1, GABBR2, IL1R2, PGR, CHRM3, IL22RA2, BMPR1B, CHRM1*
Functional Proteins	*AGR3, PPP1R14C, ROPN1, OCA2, IGFALS, IGF2BP3, INA, L3MBTL4, SFRP1, FABP7, GPRIN2, S100B, LAMP3, BPI, CNTNAP2, CRABP1, AGR2, PGLYRP2, TMEM74, TMEM150C*
Structural Proteins	*TFF1, B3GNT5, KCNK15, LOC145837, FBP1, NCRNA00092, TTC36, ABCC11, LEMD1, C10orf82, ZIC1, C8orf47, POU4F1, DNAH5, CHST4, TTYH1, CXCL1, PROM1, DLX6, ANKRD45, MIA, PYY, LPO, PCOLCE2, CHI3L1, C6orf15, UCHL1, CCDC129, PLCH1, TF, SLC6A15, CORO6, FOXG1, COCH, PTGS2, KRT4, THRSP, WIF1, C1orf64, TFF3, SLC44A4, HPX, STAC, BPIL1, FERMT1, PGBD5, IL20, LRRC17, SLC6A2, KCNS1, CXCL5, HMGCS2, ADD2, SOSTDC1, TLX1, TLX3, KLK10, SPHKAP, DSG1, ANXA8L2, IRX1, UGT2B11, C8orf46, SLC5A1, PRR15, SLC16A6, DNALI1, GSTP1, SPDEF, ST8SIA1, C1QL4, EN1, KCNE4, MUC1, C20orf103, ITGB8, CAPSL, ELF5, LIPG, SOX10, DSC3, CYP4F22, TPSD1*

**Table 3 cancers-11-01724-t003:** Ten genes targeted by specific drugs. DMS—differentially methylated CpG site.

RefGene	Region	CpG Sites	DMS	Pattern	Drugs	Drug Examples
*BPI*	TSS200	3	1	HypoUp	1	Fostamatinib
*CHRM1*	Body	3	2	HypoUp	95	Cevimeline, tramadol, succinylcholine
*CHRM3*	Body	8	6	HyperUp	75	Ziprasidone, disopyramide, ipratropium
*GABRP*	TSS200	3	2	HypoUp	16	Prazepam, quazepam, nitrazepam
	TSS1500	3	1	HypoUp	16	Prazepam, quazepam, nitrazepam
*GSTP1*	Body	6	4	HypoUp	37	Chlorambucil, cisplatin, busulfan
*PTGS2*	TSS200	3	3	HypoUp	111	Bufexamac, bendazac, acemetacin
	TSS1500	6	4	HypoUp	111	Bufexamac, bendazac, acemetacin
	Body	5	2	HypoUp	111	Bufexamac, bendazac, acemetacin
*S100B*	TSS200	2	1	HypoUp	9	Olopatadine, calcium, calcium citrate
*SLC6A2*	TSS200	3	3	HypoUp	76	Amphetamine, phentermine, tramadol
*TF*	TSS200	3	1	HypoUp	32	Cisplatin, isoflurophate, iron dextran
	TSS1500	4	2	HypoUp	32	Cisplatin, isoflurophate, iron dextran
*UCHL1*	TSS200	2	1	HypoUp	1	Phenethyl isothiocyanate

**Table 4 cancers-11-01724-t004:** Binding pattern for the PTGS2 structure (Protein Data Bank (PDB) identifier (ID): 5IKV) with their targeting drugs. FDA—Food and Drug Administration.

No.	FDA Drug	Binding Energy (kcal/mol)	No. of Hydrogen Bonds	Residues
1	Icosapent	−6.63	0	-
2	Adapalene	−6.99	1	Gln-192
3	Mesalazine	−6.84	1	Val-523
4	Dapsone	−6.48	2	Gln-192, Ser-530
5	Sapropterin	−6.15	6	Asn-87 (3), His-90, Lys-511, Glu-520
6	Flurbiprofen	−5.88	1	Gln-192
7	Ketorolac	−5.12	2	Arg-513, Val-523
8	Piroxicam	−4.71	2	Ile-517 (2)
9	Phenylbutazone	−4.28	1	Arg-513
10	Mefenamic acid	−3.55	2	Met-522, Gln-192
11	Carprofen	−3.19	1	Val-523

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
