# Peer review of "Coupled Genome-Wide DNA Methylation and Transcription Analysis Identified Rich Biomarkers and Drug Targets in Triple-Negative Breast Cancer"

_cancers, 2019, doi:10.3390/cancers11111724_

Round 1
Reviewer 1 Report
The authors submitted a new form of the manuscript according to the recommendations of the reviewers. After a thorough analysis it is clear that the manuscript is now ready to be published.
Reviewer 2 Report
It is my opinion that the authors' answers and additions have been useful in clarifying the characteristics of the TNBC and TPBC populations analysed. I encourage authors to quickly move research into in vitro and in vivo experimental evidence.
Reviewer 3 Report
In the revised version of the manuscript, the authors exhaustively addressed all the concerns made by the reviewers, thus increasing the clarity and the scientific quality of the manuscript.
This manuscript is a resubmission of an earlier submission. The following is a list of the peer review reports and author responses from that submission.
Round 1
Reviewer 1 Report
The overall paper presents a study with a very interesting and complex methodological approach.
The contents are innovative in such a way that using bioinformatics the authors present possible therapeutic targets and currently approved drugs (many of them used for other therapeutic purposes) as inhibitors of these targets.
The discussion and conclusions are very well attained with reamarkable results respecting to the use of already known pathways and drugs that are actually considered in separate but with potential to be further applied in the treatment of TNBC, a very aggresive form of breast cancer with few therapeutic solutions.
Regarding ABSTRACT:
- Line 13: change "caused" for "due"; "in distinguish" for "to distinguish".
- Line 16: change "could be" for "may"; "of identifying" for "to identify".
- Line 17: change "certain existing" for "specific".
- Line 19: change "methylation regulated" for "methylation-regulated"; add after "for TNBC" after "genes".
- Line 25: add "the" after "involved in".
Regarding INTRODUCTION:
- Line 36: add "have been" after "efforts".
- Line 37: add "despite" before "progress".
- Line 39: delete " the" before "promising".
- Line 43: delete "and" before "methylation".
- Line 45: change "could" for "may".
- Line 47: change "metastases" for "metastasis"; "change "could" for "may".
- Line 48: describe briefly what does the targeting of JAK2/STAT3 pathway (it up- or down-regulates this pathway?).
- Line 49: delete "rather"; change "limited" for "strict"; add "due to the lack" after " or".
- Line 52: change "breast cancer. For example" for "breast cancer; for example"; change "should" for "may".
- Line 56: change "could" for "may".
- Line 58: change "Certain existing drugs" for "Specific drugs may".
- Line 59: change "In the" for "In this".
- Line 60: change "mine" for "gather".
- Line 62: change "same genes, and searched for potential" for "same genes. Potential"; add "were also looked for" after "TNBC signatures genes".
- Line 63: add "considered" after "114 genes".
- Line 64: add "a method of" after "Using"; add comma after "simulation"; delete "method" after "simulation".
- Line 65: add commas after "gene" and after "(COX-2)"; change "PTGS2(COX-2)" for "PTGS2 (also known as COX-2)".
- Line 66: add commas after "drug" and after "Sapropterin".
Regarding MATERIALS AND METHODS:
- Line 69: delete "the" after "we used"; insert reference for "TCGA data"; add "both" after "downloaded".
- Line 70: delete the first word "data".
- Line 73: add comma after "(GSE78758)".
- Line 74: add comma after "database".
- Lines 81 and 82: specificy the values used for beta parameters threshold.
- Line 90: change "in this step" for "accounted for this topic".
- Line 101: "DEG" is written in Bold...use normal style.
- Line 106: "HypoUp", "HypoDown", "HyperUp" and "HypoDown". What is the measurable difference between these four distinct groups?
- Line 133: change "binding to" for "ability to bind to".
Regarding RESULTS:
- Line 141: is is stressed "statistically significant"...for what p-value? p<0,05? p<0,01? ...you must be more specific. Considered the hypothesis presenting the
results as confidence intervals as they allow a wide capacity to interpret the results once they do not rely only in p-value.
- Line 144: "hypermethylated" and "hypomethylated"...it is not clear what these terms mean...what is your threshold for characterizing a group as hyper- and another
as hypomethylated?
- Line 149: "significantly"...describe for which p-values or according to confidence intervals.
- Line 166: "KEGG" stands for? Insert reference.
- Line 169: "statistically significant"...describe for which p-values or according to confidence intervals.
- Line 221: "GSE78751" and "GSE75754"...what are those? You must describe, please.
- Line 285: change "is comprises of" for "comprises".
Regarding DISCUSSION:
- Line 311: delete "the expression of"; change "all shutdown" for "not expressed".
- Line 312: delete "their"; change "region" for "regions".
- Line 321: change "cites" for "sites".
Change "conclusion" for "Conclusion".
Reviewer 2 Report
In this manuscript, the authors analyzed epigenetic modification (DNA methylation) and gene expression as a single unit in order to identify TNBC-specific signatures targeted by existing drugs. They used TPBC cases as the control in order to provide the highest contrast to TNBC and to increase TNBC-specific signatures. They also identified 288 FDA approved drug targeting 10 of the 114 differentially methylated and expressed genes. Triple-negative breast cancer remains the most challenging BC subtype to treat and so, the identification of biomarkers that can help guide treatment decisions remains a clinically unmet need. This study may contribute to this way.
Minor comments:
Introduction. Line 40 and 41. Is the term "alternated" correct? May be “altered”? Figure 1A-C. Authors should specify (on the image or in the figure legend) the reference group (TSS1500, etc).
Major comments:
Table S1:
Table S1 reports the characteristics of TNBC and TPBC populations with n=116 and n=100 samples, respectively. Subsequent analyses of DMG and DEGs were performed on 84 samples for TNBCs and 64 samples for TPBC because of exclusion of missing data (in gene expression or methylation). Authors should report the characteristics of the effectively analyzed TNBC and TPBC populations (84 TNBCs and 64 TPBC). Box A21 mentions “TNBC subtypes” followed by Basal-like, HER”, LumA, etc. A classification that seems to refer to Breast Carcinomas subclasses (Sørlie T, et al. Gene expression patterns of breast carcinomas distinguish tumor subclasses with clinical implications. Proc Natl Acad Sci U S A. 2001 Sep 11;98(19):10869-74) instead of a TNBC classification (Lehmann BD, et al. Identification of human triple-negative breast cancer subtypes and preclinical models for selection of targeted therapies. J Clin Invest. 2011 Jul;121(7):2750-67. doi: 10.1172/JCI45014). Triple positive breast cancer (TPBC) phenotype is associated with HER2 and hormones co-expression (ER/PgR/HER-2 positive BC). In this manuscript, the TPBC population chosen by authors seems to be heterogeneous with a very low percentage (27%) of hypothetic ER/PgR/HER2-positive (among LumB subtype). In addition, a very high percentage of “unknown” phenotype has been included in the study, especially in the TNBC group. I would like to ask the authors the criteria for sample selection and TPBC discrimination. I would like to ask the authors the meaning of columns C and E “percentage (100%)”. The total count for “menopause status”, “stage” and “race” categories are inconsistent with the sum of sub-groups, for both TNBC and TPBC. For example, the sum of Peri(1)+Post(70)+Pre(19)=90 while TPBC samples are 100. Can authors give an explanation?
Reviewer 3 Report
In the manuscript entitled “Coupled genome-wide DNA methylation and transcription analysis identified rich biomarkers and drug targets in triple negative breast cancer”, Guo and colleagues show a new signature specific for TNBC. This molecular signature resulted from a combined transcriptomic/epigenetic analysis, which coupled with in silico data of already available drugs led to the identification of 356 drugs targeting 10 out of 114 dysregulated genes.
The following are the concerns of this reviewer:
It is not clear to this reviewer why the healthy tissue has not been included in the study, to be sure the identified target and relative drug will not affect healthy counterpart in breast cancer patients. The lack of molecular/functional validation of the reported findings is a minus for the research here presented. It is not clear what the classification of triple positive breast cancer would include. The “non TNBC” class includes luminal A, luminal B, HER2 enriched, which are characterized by specific behavior and progression. It seems inappropriate to pool all this breast cancer molecular subset together. Moreover, it would have been more useful for clinical settings to identify the metastatic “non TNBC” to treat patients that are now undergoing standard therapy thus evolving into progressive disease. It is not clear how many of the identified drugs are specifically targeting the dysregulated genes/pathways. It is not clear why none of the identified drugs have been tested in vitro to demonstrate at least an induction of cell death in TNBC cell line and healhty counterpart.